# Detection of the Cyclic Imines Pinnatoxin G, 13-Desmethyl Spirolide C and 20-Methyl Spirolide G in Bivalve Molluscs from Great Britain

**DOI:** 10.3390/md22120556

**Published:** 2024-12-12

**Authors:** Ryan P. Alexander, Alison O’Neill, Karl J. Dean, Andrew D. Turner, Benjamin H. Maskrey

**Affiliations:** Centre for Environment Fisheries and Aquaculture Science (CEFAS), Barrack Road, Weymouth DT4 8UB, UKkarl.dean@cefas.gov.uk (K.J.D.); andrew.turner@cefas.gov.uk (A.D.T.); ben.maskrey@cefas.gov.uk (B.H.M.)

**Keywords:** cyclic imines, pinnatoxins, brevetoxins, bivalve molluscs, harmful algae, LC-MS/MS, emerging toxins, fatty acids, esters, LC-HRMS

## Abstract

Harmful algal biotoxins in the marine environment are a threat to human food safety due to their bioaccumulation in bivalve shellfish. Whilst official control monitoring provides ongoing risk management for regulated toxins in live bivalve molluscs, no routine monitoring system is currently in operation in the UK for other non-regulated toxins. To assess the potential presence of such compounds, a systematic screen of bivalve shellfish was conducted throughout Great Britain. A rapid dispersive methanolic extraction was used with UHPLC-MS/MS analysis to test for fifteen cyclic imines and seven brevetoxins in 2671 shellfish samples taken from designated shellfish harvesting areas around Great Britain during 2018. Out of the 22 toxins incorporated into the method, only pinnatoxin G, 13-desmethyl spirolide C and 20-methyl spirolide G were detected, with maximum concentrations of 85.4 µg/kg, 13.4 µg/kg and 51.4 µg/kg, respectively. A follow up study of pinnatoxin G-positive samples examined its potential esterification to fatty acids and concluded that following hydrolysis, pinnatoxin G concentration increased by an average of 8.6%, with the tentative identification of these esters determined by LC-HRMS. This study highlights the requirement for ongoing monitoring of emerging threats and the requirement for toxicological and risk assessment studies.

## 1. Introduction

Shellfish are a globally important group of organisms that are both a traditional food source, with evidence of their use as sustenance dating back 125,000 years [1], and they are arguably the world’s most sustainable source of animal protein [2,3]. Considering this and other factors, the global shellfish industry has grown year on year, with shellfish species comprising 52.2% of the cultured aquatic species in United Nations countries, reaching a global revenue of USD 104.55 billion in 2018 [4,5]. However, socio-economic risks and related challenges exist in the industry as a result of the ability of molluscs to accumulate environmental contaminants, which has led to strict controls and regulations on the industry to ensure food safety and quality. One of these challenges concerns the uptake and storage of harmful algal biotoxins via filter feeding, which can be harmful to both human and animal health [6]. Various toxin groups have been identified globally, among these, three groups are regulated (Amnesic Shellfish Toxins (AST), Lipophilic Toxins (LT) and Paralytic Shellfish Toxins (PST)) which are known to be produced by a range of microalgal species. EU legislation consequently dictates the need for official control (OC) monitoring of classified shellfish harvesting areas for regulated marine biotoxins in bivalve mollusc flesh and their causative microalgal species in seawater [7,8]. However, there is the potential for other microalgal species and their associated toxins to spread to new locations including Great Britain (GB), creating an emerging threat for human and animal health [9,10,11]. Two of the emerging toxin groups previously identified as potential risks within GB are Cyclic Imines (CIs) [11] and Brevetoxins (BTXs or PbTxs) [10].

Currently the marine biotoxin group known as CIs [12] comprises sixteen spirolide (SPX) analogues [11], five gymnodimine (GYM) analogues [13], eight pinnatoxin (PnTx) analogues [11], three pteriatoxin (PtTX) analogues [11,14], two prorocentrolide analogues [11,15,16], one spiro-prorocentrimine analogue [17] and portimine [18]. CIs are grouped together according to both their chemical structure, which contains an imino group pharmacore and spiro-linked ether moieties [12,19,20], as well as their similar toxicological properties [21]. Globally, no regulatory limits have been set for these toxins in shellfish [19]; however, human health guidelines have been published with the following suggested limits: at 0.4 mg SPX/kg for SPX [22], and for PnTxs, both 23 µg PnTx G/kg [23] and 268 µg PnTx G/kg [24]. CIs are currently not associated with human poisonings [19], although a recent *Vulcanodinium rugosum* bloom was linked to a dermatitis outbreak in Cuba [25]. Portimine, PnTx E and PnTx F were present in the bloom, but it is unknown whether these compounds were responsible.

BTXs comprise a range of analogues produced by both causative phytoplankton as well as shellfish metabolites, including PbTx-1-3, PbTx-5-10, BTX-B1-B5 and S-desoxy-BTX-B2 [10,26,27]. These are complex, polycyclic ethers that are split into two main groups, those with either ten (A) or eleven ether rings in their backbone structure (B) [26,27]. BTXs can accumulate in shellfish tissues, and following ingestion, they may cause a range of gastrointestinal and neurological symptoms [28] termed as Neurotoxic Shellfish Poisoning (NSP) [26]. The toxins can also negatively impact human health through the inhalation of toxic aerosols via airborne pathways [29,30]. BTXs have had a large impact on animal health, including widespread fish and bird kills through ‘red tide’ blooms [6,26,27,31]. The 2019 National Shellfish Sanitation Program (NSSP) Guide for the Control of Molluscan Shellfish produced by the US Food and Drug Administration (FDA) and the US Environmental Protection Agency (EPA) states that a shellfish harvesting site’s status (open or closed) is to be set to closed when an NSP outbreak with a limit of 0.8 mg brevetoxin-2 equivalents/kg is reached in shellfish tissues [32]. In respect to regulatory limits, toxin concentrations are often expressed as toxic equivalents/kg due to differences in toxicity between analogues (i.e., PnTxs). Toxic Equivalency Factors (TEF) provide the relative toxicity of an analogue in relation to the parent toxin and are used to convert the concentrations of a toxin into equivalents/kg, thus giving an indication of overall toxicity rather than just the concentration. Calculating the relative toxicities of analogues relies on toxicological information that is often lacking.

CI and BTX production in the marine environment are predominantly attributed to dinoflagellates. SPX is produced by *Alexandrium ostenfeldii* [33] and, according to some studies, *Alexandrium peruvianum* [34,35]. It should be noted that the two species are sister species that are very hard to differentiate; they are morphologically very similar and are often used as synonyms of each other, despite claims that they are separate species and exhibit differences [36,37,38]. Some even consider them to be heterotypic synonyms of each other, stating the ‘classification of *A. peruvianum* should be discontinued until alternative criteria are available’ [39] (and henceforth, when either species is mentioned, they will be referred to as the same species). The production of GYM A, B and C is attributed to *Karenia selliformis* [40,41], whilst the production of the other analogues, 12-methyl GYM, 12-methyl GYM B and GYM D, are produced by *A. ostenfeldii* [13,35,42]. PnTxs are produced by just one known species to date, *V. rugosum* [43,44]. The source of pteriatoxins has not been identified to be the product of a specific dinoflagellate species; however, it is hypothesised that it is produced via the biotransformation in shellfish [45] after three toxins were found in the bivalves *Pteria penguin* [14]. The full distribution of prorocentrolides is unknown, but *Prorocentrum lima* is responsible for the production of prorocentrolide A [16] and prorocentrolide B by *Prorocentrum maculosum* [15]. Spiro-prorocentrimines are hypothesised to be produced by *Prorocentrum* spp. [17]. Portimine is produced from the same dinoflagellate species that produces the PnTxs *V. rugosum* [18]. BTXs are produced by the dinoflagellate *Karenia brevis* [46] and are also linked to the raphidophyte species *Chattonella antiqua*, *Chattonella marina*, *Heterosigma akashiwo* and *Fibrocapsa japonica* [27]. Raphidophyte BTX production however is reported to be ichthyotoxic and has little effect on human health, unlike the blooms formed by *K. brevis* [27].

Spirolides and their associated dinoflagellates have previously been detected in GB, specifically in Scotland [36,47,48,49]. *A. ostenfeldii* has been the sole causative algae found in Scottish waters producing a range of different spirolides. It has been shown to solely produce 20-methyl spirolide G [36] whilst also producing spirolide A, desmethyl spirolide C and spirolide C [47,48] in varying concentrations at different locations. They have also been detected in Europe, in Denmark [50], France [51], Ireland [34,52], Italy [53], the Mediterranean [54], Norway [55,56], Portugal and Slovenia [57] and Spain [57,58]. Outside of Europe they have been detected in Argentina [59], Canada [60,61], Chile [62], China [63], Mexico [64], New Zealand [9] and the USA [65]. GYM and their associated dinoflagellates have not yet been detected in GB waters. In 2015, GYM was recognised as having not been detected in European waters [11]; however, within the same and the following year, this was contradicted with its discovery in the Baltic Sea [13,54], a first for Europe. Internationally, however, GYM has been found in Australia [66], China [63], Mexico [64], New Zealand [67], South Africa [68] and Tunisia [69]. PnTx, portimine and their associated dinoflagellates have currently only been detected in Scotland [49], as well as in Denmark, Netherlands, Portugal and Italy [57,70], France [43,71], Ireland [52], Norway [55] and Spain [58] in Europe. They have also been detected in Canada [72], Japan [73], Mexico [64], New Zealand [9,18,74] and South Australia [45,75]. Pteriatoxins have not been detected in GB or Europe [19] and have only been detected in the bivalves *P. penguin* [14] in Japan. To date, prorocentrolides, spiro-prorocentrimines and their associated dinoflagellates have not been reported in shellfish in GB or Europe [19]. They have only been reported internationally in Canada [61], Japan [16] and Taiwan [17]. Finally, BTXs and their associated dinoflagellates have not been found in GB or European waters until recently [6,10,27], where BTX-2 and BTX-3 were detected in France [76]. They have only been found internationally in New Zealand [28] and the USA [31]. It is hypothesised that with the right environmental conditions, BTX presence in GB will be observed in the future [10,77], especially with upcoming climate change-induced changes to the environment.

In this study, 2671 shellfish samples were examined from official designated shellfish harvesting areas around England, Wales and Scotland from 1 January 2018 to 31 December 2018. UHPLC-MS/MS and LC-HRMS were used to detect and quantify concentrations of CIs and BTXs (as detailed in Table 1 and Figure 1) in GB shellfish species.

## 2. Results

### 2.1. Toxin Detection

Of the 22 toxin analytes determined in the 2671 samples, PnTx G, SPX1 and 20-Me-SPX-G were detected above their respective LOQs of 0.06, 0.06 and 0.05 µg/kg (LOQs obtained from [82]). PnTx G was detected in 733 samples (27.4%), SPX1 in 181 samples (6.8%) and 20-Me-SPX-G in 1106 samples (41.4%). The maximum concentrations were 85.4 µg/kg, 13.4 µg/kg and 51.4 µg/kg for PnTx G, SPX1 and 20-Me-SPX-G, respectively. PnTx A and PnTx H peaks were detected below the LOQ. Refer to Section 4.4 for the specific LOQs of each toxin.

Figure 2 displays some example of Multiple Reaction Monitoring (MRM) chromatograms for the PnTx G, SPX1 and 20-Me-SPX-G present in both a calibration standard and in the selected shellfish extracts.

### 2.2. Toxin Profiles

Toxin profiles were obtained by normalising the concentration of each toxin within every sample relative to the total concentration in each sample. These were then averaged over each month and the whole year. As shown in Figure 3, the average annual toxin profile across all samples contained 37.9% PnTx G, 5.8% SPX1 and 56.3% 20-Me-SPX-G. The lowest percentage of toxins detected in the profile occurred in May for PnTx G (13.3%), August for 20-Me-SPX-G (41.5%) and April, September and November for SPX1 (0%). The highest percentage of toxins detected in the profile occurred in August for PnTx G (56.8%), May for 20-Me-SPX-G (84.6%) and January and February for SPX1 (22.9%).

### 2.3. Spatial Analysis

Table 2 shows that 49 out of the 53 sublocations sampled (subdivision within the broader geographic regions) contained one or more of the emerging toxins PnTx G, SPX1 or 20-Me-SPX-G at detectable concentrations above the LOQ. The geographic region containing shellfish with the maximum concentration of PnTx G was Central–West Scotland. The highest percentage of PnTx G-positive samples relative to the amount tested, however, was in Northeast Scotland, at 63.2%. Samples from three areas had concentrations of PnTx G over the proposed 23 µg/kg human health guidance limit; Central–West Scotland had 30 samples over the human health guidance limit, Southwest England had two and Shetland in Northeast Scotland had 36. The maximum concentration of SPX1 was in Southwest England, although only one sample contained detectable SPX1 at this site. The highest percentage of positive samples tested for SPX1 was located in Northeast England, at 35.7%. Samples from this region also had the maximum concentration of 20-Me-SPX-G detected and that was also the joint highest along with Southwest England, with 100% of the samples tested being positive for 20-Me-SPX-G. It should be noted, however, that only one sample was tested from this region in England.

The statistical analysis of the toxin concentrations revealed that both PnTx G and 20-Me-SPX-G concentration had a significant relationship with location, highlighting spatial variability in concentration. SPX1 concentration was not statistically significant when compared with other locations. Interpretation of the epsilon squared value shows that location has a large effect on PnTx G and SPX1 concentration and medium effect size on 20-Me-SPX-G concentration.

### 2.4. Temporal Analysis

Table 3 summarises the differences between the month and the concentrations of the three toxins. Both PnTx G and 20-Me-SPX-G were detected every month. The maximum concentration of PnTx G was observed in July, SPX1 in August and 20-Me-SPX-G in June. The months were split into seasons (see Appendix A) to see if any statistical relationships existed, with the statistical relationships resembling those observed when split into months. One interesting finding was that the highest count of 20-Me-SPX-G-positive samples was in the summer, at 388, but the highest percentage of 20-Me-SPX-G-positive samples was in the winter, at 50.9% of the total tested.

Kruskal–Wallis with Dunn pairwise output (see Appendix A) revealed that all three toxins had a statistically significant relationship with month. This potentially represents the months in which the source algae are blooming. Month had a medium effect size on PnTx G and SPX1 concentration but only a small effect size on 20-Me-SPX-G. Kruskal–Wallis with Dunn pairwise output was also similiar when months were grouped by season (see Appendix A), with all three toxins having a statistically significant relationship with season. Season had a medium effect size on SPX1, and a small effect size on both PnTx G and 20-Me-SPX-G.

### 2.5. Bivalve Species

Table 4 summarises the differences between species and toxin concentrations. Mussels (*Mytilus* spp.) were the most sampled species with 1724 samples tested, with 86.5% of these being positive for PnTx G, SPX1 and/or 20-Me-SPX-G. Every species, except queen scallops (*Aequipecten opercularis*) (*n* = 3), had at least one of the toxins detected, with cockles (*Cerastoderma* spp.), mussels and Pacific oysters (*Magallana gigas*) presenting with all three toxins. PnTx G was detected in 69.3% of positive mussel samples, 20-Me-SPX-G in 96.8% of positive Pacific oyster samples and SPX1 was detected in 100% of positive hard clam (*Mercenaria mercenaria*), king scallop (*Pecten maximus*) and manila clam (*Ruditapes philippinarum*) samples. It should be noted that positive samples were low for these species; *n* = 2, 2 and 3, respectively.

Kruskal–Wallis with Dunn pairwise output (see Appendix A) revealed that PnTx G and 20-Me-SPX-G had a statistically significant relationship with species. SPX1 was not statistically significant, meaning that although multiple species contained SPX1, no species was statistically different from another in the concentrations detected. Species had a very small effect size on PnTx G concentration, a medium effect size on SPX1 and a small effect size on 20-Me-SPX-G.

### 2.6. Analysis of Group Differences with Post Hoc Comparisons

Table 5 highlights that location has the highest effect on the concentrations of all the toxins detected. If a variable is statisitically significant from a toxin, it signifies that one or more factors within a variable differ significantly from another.

### 2.7. PnTx G Esters

Previous studies have highlighted the potential for esterification of PnTx G with fatty acids [72,83]. In order to investigate this, an additional 21 samples containing high free PnTx G concentrations were subjected to alkaline hydrolysis and the concentrations of PnTx G were determined in both unhydrolysed and hydrolysed samples to give the ‘free’ and ‘total’ concentrations, respectively. The hydrolysed concentration of PnTx G increased in all samples compared with the unhydrolysed concentration (see Appendix A), ranging from 1.7% at the lowest increase up to 29.0% at the highest increase and with a mean total difference of +8.6%. This suggests the presence of PnTx G esters and is a potential indicator of the underestimation of the PnTx G concentration in the main study (hydrolysis was not undertaken).

### 2.8. Identification of PnTx G Esters

To further investigate the presence of PnTx G esters, two samples containing the highest difference in concentration (Appendix A; samples three and eight) were analysed for PnTx G fatty acid esters via LC-high resolution mass spectrometry (LC-HRMS). The chromatographic profiles of the theoretical masses of the fatty acid esters of PnTx G are displayed in Figure 4, with findings on the exact mass, calculated mass, mass error, retention time and peak area for each ester displayed in Table 6. In total, 14 out of 26 potential fatty acid esters were detected in the unhydrolysed extracts of both samples. With the exception of 22:2 and 22:5, all esters demonstrated a mass accuracy equivalent to <2 ppm, suggesting good confidence in identification. To further evidence their identity as fatty acid esters, none were detected when analysed post-hydrolysis. Fatty acid 17:1 was found to be the dominant ester accounting for the total peak area at 54.25%, with only 18:1 and 19:0 encompassing more than 10% of the total peak area. The remaining lipids were all below 10%, with the assumption that all compounds have a comparable ionisation efficiency. Two esters were found only in sample three, 18:0 and 20:5, and one ester was found only in sample eight; 22:2. These two samples were taken from the same location and sampled seven days apart, which could potentially be a reflection of natural ester change over time. However, as all three of these lipids are below 1% of the total peak area, these changes over time are very small and not necessarily indicative of what is happening in the environment.

## 3. Discussion

### 3.1. Main Findings

This is the first study showcasing the large distribution and concentration of the three toxins, CIs PnTx G, SPX1 and 20-Me-SPX-G, in shellfish species harvested in GB. PnTX G has previously been detected in shellfish in Scotland [49] along with the detection of SPX1 and 20-Me-SPX-G in the water column in Scotland [36,47,48], so their presence in shellfish tissue was expected; however, the breadth and depth of this study increases the understanding of the global distribution and geographic range of these toxins, with their expansion across GB from mainland Europe. Within Europe, PnTx G distribution is relatively extensive, found in countries such as Denmark, Netherlands, Portugal and Italy [57], France [43,71], Ireland [52], Norway [55] and Spain [58].

The concentrations of PnTx G in 68 samples from this study exceeded the human health guidance limit of 23 µg PnTx G/kg suggested by Arnich and colleagues [23]. However, a study published in 2024 [24] alternatively suggested a concentration of 268 µg PnTx G/kg is safe for human consumption. This subsequently means that all of the shellfish sampled within this study are below this human health guidance limit. The concentrations of PnTx G in this study are in line with the concentrations observed in shellfish in Scotland [49], Canada [72] (Northwest Atlantic), Norway [55] (Northeast Atlantic) and Spain [58] (Northwest Mediterranean), higher than concentrations observed in other North Atlantic countries in Europe [57], Spain [84] and Ireland [52]. This is interesting given the geographical proximity of Ireland to GB, with lower concentrations reported in shellfish from New Zealand [9] and France [71] (Mediterranean). Although sample numbers in some of the previous studies were low, some exhibited a higher number of samples positive for PnTx G and some were taken from different water bodies. These results suggest that the presence of PnTx G varies by location, both within and outside countries, and it is heavily influenced by environmental factors that affect bloom dynamics and the uptake of toxins in shellfish. The wide geographic sampling net of this study fully captures the background and maximum concentrations of PnTx G seen over a one-year period across GB.

The concentrations found in this study for both SPX1 and 20-Me-SPX-G were considerably lower than the advised human health guidance limit of 0.4 mg SPX/kg [22]. Most of the samples positive for SPX1 (69%) were close to the LOQ and below 1 µg/kg. Moreover, 51% of the samples positive for 20-Me-SPX-G were still below 1% (4 µg/kg) of the advised human health guideline limit. The concentration of SPX1 and 20-Me-SPX-G in this study is similar to levels seen in samples collected from Spain (Northwest Mediterranean) [58], higher than levels observed in Ireland [52], and lower than samples from Europe [57], France (Atlantic coast) [51] and Norway (North Atlantic) [55]. Although sample numbers in some of the previous studies were low, some exhibited a higher number of samples positive for SPX1 and 20-Me-SPX-G, and some were taken from different water bodies. These results strongly imply that the presence of SPX1 and 20-Me-SPX-G is variable by site locations both within and outside countries and are heavily influenced by environmental factors that affect bloom dynamics and the uptake of toxins in shellfish.

### 3.2. Spatial Variability

PnTx G was found primarily in two places, Northeast Scotland and Middle West Scotland. Southwest England and South Wales were the only places outside of Scotland to contain any levels of PnTx G; 16% and 77% of the samples were positive for PnTx G, respectively. This is significant because it implies that PnTx G producers such as *V. rugosum* proliferate more along the Scottish coast and Southwest Atlantic compared with the southeast coast and the channel. The lowest percentage of PnTx G-positive samples was in Northwest Scotland (8%), despite extensive sampling in the area over the one-year period, suggesting the presence of a natural environmental barrier that prevents PnTx G producers from proliferating into a bloom in that area [85]. Despite this, this yearlong study can be assumed to capture a snapshot of the potential background and maximum concentrations that could potentially be seen across GB from year to year. Seawater temperature, NH^4+^, wind speed, light intensity, nutrient concentration, salinity and PO_4_^3−^ have been proven to impact both PnTx G concentration in shellfish and *V. rugosum* presence [86], which all change depending on specific locations, with many studies demonstrating that PnTx G concentration can vary both spatially and temporally [55,58,71,84,85].

SPX1 was comparable across sites, with a low/similar positive count at each location, whilst Northeast England demonstrated a significant detection of 20-Me-SPX-G when compared with other areas. The toxin concentrations of SPX1 and 20-Me-SPX-G at each location in this study cannot be compared with those previously obtained from Scotland as no toxin concentration was extrapolated in those prior studies, only cell count and abundance. However, it does showcase that *A. ostenfeldii* is not restricted to just Scotland [11,36,47,48], and toxins at each location are likely not a problem for human health at the concentrations seen in the shellfish tissue harvested [19,22].

### 3.3. Temporal Trends

Month had a statistically significant effect on PnTx G concentration, with a higher level observed during the hotter summer months and a lower level observed during the colder months. This suggests the bloom of *V. rugosum* or unknown PnTx G producers around the summer/early autumn, leading to the accumulation of PnTx G in shellfish tissue. PnTx G then slowly depurated out of the shellfish tissue or was broken down by April, before the cycle began again in the summer. Our data align with the previously observed natural bloom dynamics for *V. rugosum*, where the highest abundances were observed in the summer months between June and September [85], reinforcing the idea that the toxins observed are due to blooms that then slowly depurate over the winter months. Studies conducted in France (Mediterranean) [71,86] showed a similar seasonal variation to the current study, whilst a study in Spain (Atlantic) [84] indicated that PnTx G followed a seasonal pattern in which the maximum concentrations took place in the winter. Although sampling was undertaken in different water bodies in some of the previous studies, these results suggest that the strain of *V. rugosum* responsible for the toxins observed in this study could be more closely aligned to the one found in France than the one found in Spain, by being a species complex [87]. A longer-term study in combination with phytoplankton analyses is therefore required.

When comparing the month and season of SPX1 and 20-Me-SPX-G concentrations, a similar trend is expected due to them being produced by the same species (*A. ostenfeldii*). However, the only similarity between the two spirolides is that they both show the highest average and maximum concentrations in summer, which is assumed to be due to environmental conditions in the summer that encourage growth [88,89,90]. The seasonal patterns observed for 20-Me-SPX-G concentration in this study are similar to a study conducted in Norway (Northeast Atlantic) [55]. The lower concentration but higher detection of SPX1 for the period from January to March compared with May to August indicates low background levels in the winter and small blooming events in the summer. In Southwest Finland (Baltic Sea, Northeast Atlantic), *A. ostenfeldii* was detected between the months from May through to September; however, *A. ostenfeldii* only bloomed between July and August, and this was dependent on location [88]. The seasonal peaks and troughs on either side of the summer period could be explained by the low levels before and after a bloom of *A. ostenfeldii*. Blooms of *A. ostenfeldii* in the Netherlands (Northeast Atlantic) [89] over a 3-year period started in June and lasted anywhere from 1 to 3 months; evidence of proliferation at warmer temperatures during the summer months in a nearby country (Northeast Atlantic) would help explain why both spirolides occur at higher concentrations during the summer. The constant detection of 20-Me-SPX-G throughout the year could be explained by the ability of 20-Me-SPX-G to slowly depurate/stick to shellfish tissue; this is implied in a study from Norway (Northeast Atlantic) [55], where 20-Me-SPX-G was detected in colder months in the absence or low abundance of *A. ostenfeldii* in the water. This does not explain why an absence of SPX1 during the autumn is observed; however, a hypothesis is that SPX1 potentially depurates from shellfish tissue at a much quicker rate. This temporal difference between the two toxins could also theoretically be explained by the slightly different source populations of *A. ostenfeldii*; these species are morphologically and physiologically very similar but produce different toxins which can co-exist or live in separate locations [55,90]. A study over the summer months in Ireland (closest North Atlantic neighbour) [91] found that levels of SPX1 were generally low, with low *A. ostenfeldii* cell counts; however, the maximum concentration was observed in June (2007) and late July/August (2008) and is hypothesised to occur alongside other *Alexandrium* spp. via synchronous germination [91]. These results suggest the occurrence of summer blooms of *A. ostenfeldii* which leads to higher concentrations of SPX1 and 20-Me-SPX-G. This is followed by the fast depuration of SPX1 and the slow depuration of 20-Me-SPX-G in the tissues of shellfish throughout the year.

### 3.4. Species

In this study, PnTx G was only present in three species, mussels, cockles and Pacific oysters. An interesting observation is that Pacific oysters had a smaller percentage of samples positive for PnTx G when compared with cockles, despite a similar mean concentration of 4 µg PnTx G/kg. The variable ability of different species to accumulate toxins is due to both physiological and ecological niche differences [71]. The propensity of mussels to contain higher concentrations and positive results for PnTx G is in line with previous studies in the Atlantic [84] and Mediterranean [58] coasts of Spain. With respect to food safety, this study demonstrates that mussels would be the ideal sentinel species for PnTx G due to their ability to accumulate more toxins; this assertion is supported by the findings in [71]. Mussels are harvested from both wild and farm locations, and it has been shown that wild populations from the intertidal zone retain higher concentrations of PnTx G when compared with raft-cultured mussels, suggesting that the toxin-producing organisms preferentially develop in shallow areas [84]. This point should be considered in the future, especially with respect to monitoring and the climate change-driven movements of PnTx G and its producers to GB waters.

SPX1 was present in the majority of species, despite the low number of positive samples in all but mussels, cockles and Pacific oysters; interestingly, the specific species PnTx G was detected. Concentration is also low across the board, with only mussels, native oysters, razor clams and surf clams having a concentration >1 µg SPX1/kg. This is in line with a study in Spain (Mediterranean) [58] where SPX1 was detected in both mussel and Pacific oyster samples at similar concentrations to the current study. These results suggest that the SPX1 concentration is in line with those detected previously in Europe.

Moreover, 20-Me-SPX-G was observed in six species out of the ten tested, including mussels. The concentrations in mussels detected in this study are in line with the concentrations found in Norwegian mussels (Northeast Atlantic) [55,56]. Although 20-Me-SPX-G-positive samples were higher in some of the previous studies, these results suggest that the 20-Me-SPX-G concentration is in line with those detected previously in Europe.

### 3.5. Non-Detected Toxins

The non-detection of 19 analytes incorporated into our methods implies the following two key points: (1) There are no sources (algal producers) for these non-regulated toxins in GB waters, and (2) if there are algal sources for these non-regulated toxins, environmental conditions over the year did not favour their growth to high enough levels to observe them in shellfish tissue. Another interesting finding is that PnTx A and PnTx H were detected at non-quantifiable traces, and we did not see any peaks for PnTx B-C or pteriatoxins A-C, which are a by-product of the biotransformation of PnTx G in bivalves [45]. With the high presence of PnTx G observed in this study, at least one of these analogues was expected to be at a concentration high enough to be quantified. For instance, PnTx A was found at a concentration of 2% or lower of the total PnTx G concentration in mussels and clams on the Mediterranean coast of France [71], whilst pteriatoxins were present at non-quantifiable traces. The absence of PnTx A and pteriatoxins could simply be explained by the difference in water bodies; the Mediterranean is associated with different climatic conditions, including higher water temperature. It is assumed that in the current study, the shellfish harvested were not efficient at breaking down PnTx G or PnTx G by-products were at undetectable concentrations, with the latter being more likely. The absence of PnTx E and D was expected, as these are by-products of PnTx F [45], which was also absent from the study. *K. selliformis* is associated with the production of GYM A, B and C [40,41], and so the absence of these toxins implies the absence of *K. selliformis* in the sampling areas, or at least the absence of high enough cell concentrations to cause toxin retention in shellfish tissue. *A. ostenfeldii* is also associated with the production of GYM A, B and C [54], as well as 12-methyl GYM B and GYM D [13,42], whilst *A. peruvianum* has been associated with the production of 12-methyl GYM [35]. Spirolides such as SPX1 are closely related to GYM due to their structure [35] and were thus expected to be detected due to the presence of SPX1 and 20-Me-SPX-G. An assumption without evidence suggests that the species responsible for producing the toxins in this study are not those referenced above; it has been shown that species can alternate the toxins they produce depending on where they are found [37,54], implying that subspecies differ on a genetic level [87] compared to species found in different geographical areas. No BTXs were found in this study; however, the first detections of BTX-2 and BTX-3 were recently found along the Mediterranean coast of France [76], alongside GYM-A detection at both Mediterranean and French Atlantic coasts. This has outlined future needs to continue monitoring for emerging unregulated toxins in GB waters that could have an effect on future food safety.

### 3.6. Esterification of PnTx G

The importance of fatty acid esters and toxicity in GB shellfish has been shown previously for regulated toxins such as Okadaic acid [92], and non-regulated toxins like PnTx G [72,83] and portimine [93], a CI derived from the same source algal species as PnTx G (*V. rugosum*). PnTx A fatty acid esters have not, to the authors knowledge, ever been directly detected; however, increases in concentration have been observed after hydrolyses [72]. Analysis of unhydrolysed vs. hydrolysed PnTx G samples demonstrated that concentrations could increase anywhere from 1.66% to 29.06%. If the higher percentage is translated onto the maximum PnTx G concentration observed in 2018, it potentially implies the maximum concentration of 85.4 µg/kg could have been as high as 110.22 µg/kg. This finding is similar to that found in eastern Canada (Northwest Atlantic) [72] and at the Ingril lagoon, France (Mediterranean) [83], where strong evidence of PnTx G fatty acid ester metabolites was found. This potential increase in the concentration and presence of fatty acid esters should be considered for future food safety regulations regarding PnTx G and other CIs.

LC-HRMS has previously been used to detect PnTx G fatty acid esters [72], whereas, in another study, LC-HRMS was unable to detect any fatty acid esters of PnTx G in the Ingril lagoon, France (Mediterranean) [83], with the successful detection of esters only obtained using LC-MS/MS instrumentation. Therefore, the potential to detect all fatty acid esters of PnTx G may be restricted in this study by the sole use of LC-HRMS.

This highlights that PnTx G has the potential to become esterified in shellfish, and that future analyses of PnTx G should include a hydrolysis step to liberate and quantify any esterified toxins in the context of food safety.

### 3.7. Source of Production

The potential source of PnTx G around the English, Scottish and Welsh coastlines has not been investigated in this study, but it suggests either the presence of PnTx G-producing *V. rugosum* or unknown PnTx G producers in GB waters. The toxins detected in this study suggest that the causative species is likely a PnTx G producer only, with similar toxin recovery from the European/Mediterranean variants of *V. rugosum* in Spain [58,84] and France [71]. It is unlikely that the *V. rugosum* variant is from New Zealand, as it produces a toxin profile with PnTx D, E and F over G [9,45,75]. Coupling this knowledge with the assumption that *V. rugosum* is potentially a species complex [87] and the lack of phytoplankton analysis within the study, a more thorough analysis should be carried out on PnTx G producers along the GB coastline. It should be noted that other toxins, such as PnTx A and H, which were below the LOQ may have been missed by this study and could theoretically be present in low concentrations.

The potential source for the spirolides could be *A. ostenfeldii*, which has previously shown the ability to produce spirolides within the Atlantic, with SPX1 production observed in Canada [33] and Scotland [47] and 20-Me-SPX-G in Scotland [36], or produce SPX1 in Ireland [34] and the U.S.A [35,94] as the synonym *A. peruvianum*. The results from this study highlight that *A. ostenfeldii* may have a much broader range than originally thought across the GB coastline, being sporadically detected in Scottish waters [36,47] and Irish waters [34], and with a distribution that extends down to the south of England in Falmouth [95]. Despite being synonyms of each other, the literature has typically described *A. ostenfeldii* as a cold-water species whilst *A. peruvianum* lives in warmer environments [95,96] and so, theoretically, the differences in toxins could be explained by the differences in temperature. However, as neither temperature nor phytoplankton were studied, this should be viewed as conjecture. The community assemblages seen in every region would need to be investigated in a future study that would help elucidate the causative organism, especially as no classification of algal species was undertaken. To the authors’ knowledge, there is no other SPX1 or 20-Me-SPX-G producer globally. Assumptions on the causative species in this study have used previous research concerning both *A. ostenfeldii* and *A. peruvianum* due to their synonymous relationship.

## 4. Materials and Methods

### 4.1. Chemicals, Reagents and Solvents

HPLC- and LC-MS-grade methanol was acquired from Fisher Scientific (Loughborough, UK). LC-MS-grade water was produced by an in-house MilliQ water purification system (Merck group, Darmstadt, Germany). Ammonium fluoride (>98% purity) was acquired from VWR International Ltd. (Lutterworth, UK). Certified toxins for working standards were acquired from the Institute of Biotoxin Metrology, National Research Council (NRC, Halifax, NS, Canada), MARBIONC, (Wilmington, DE, USA), Cawthron Natural Compounds, (CNC, Nelson, New Zealand) and CIFGA (Lugo, Spain).

### 4.2. Samples

2671 shellfish samples (cockles (*Cerastoderma* spp.); Mussels (*Mytilus* spp.); Queen scallops (*Aequipecten opercularis*); Pacific oysters (*Magallana gigas*); Hard clams (*Mercenaria mercenaria*); King scallops (*Pecten maximus*); Manila clams (*Ruditapes philippinarum*); Razor clams (*Ensis* spp. and *Solen* spp.); Native oysters (*Ostrea edulis*); Surf clams (*Spisula* spp.)) were obtained between 1 January 2018 to 31 December 2018 from the Official control biotoxin monitoring programmes and used with kind permission from Food Standards Scotland (FSS) and the Food Standards Agency (FSA). Samples were shucked and homogenised upon receipt and sent for extraction. Homogenisation was undertaken when the sample contained ten or greater individuals that were alive and had a total flesh weight >50 g.

For analysis of PnTx G esterification, samples were obtained between May–October 2023 from the Scottish regulatory monitoring programme. Samples were selected based on their PnTx G concentration around the human health guidance limit of 23 µg PnTx G/kg [23].

### 4.3. Sample Extraction

Extracted samples were obtained from the English, Scottish and Welsh harmful algal biotoxin official control monitoring programmes following the EU-harmonised Standard Operating Procedure method [97]. This was achieved using a double exhaustive extraction through the addition of 1 × 9 mL methanol to 2.0 ± 0.01 g of homogenised shellfish tissue, vortex-mixing for three minutes at 2500 rpm, centrifuging for eight minutes at 20 °C and then pouring off the supernatant. Another 9 mL methanol was then added to the pellet which was then subsequently blended with a T 25 digital Ultra-Turrax (IKA, Oxford, UK) for one minute and then re-centrifuged with the same settings as before. The supernatant was then combined with the first to give a single sample and subsequently topped up to 20 mL with methanol. The supernatant was then filtered into a vial using a Millipore 0.2 µm nylon syringe filter and then stored in a −20 °C freezer until analysis by UHPLC-MS/MS. Analysis was undertaken between December 2019 and June 2020 in large batches to reduce variability.

For the analysis of PnTx G esterification, all the samples selected had already been extracted following a procedure similar to [97], with the only difference being that samples were extracted in 3 × 6 mL methanol extractions with the use of a vortex mixer on each extraction instead of with a T 25 digital Ultra-Turrax. These extracts were then kept in the −20 °C freezer until analysis in November 2023. A total of 1 mL of extracts were hydrolysed with 125 µL 2.5 M NaOH and allowed to heat for 40 min at 76 °C and then neutralised with 125 µL 2.5 M HCl.

### 4.4. Sample Analysis

Sample analysis was performed using a Waters (Manchester, UK) Acquity I-Class UHPLC system coupled to a Xevo-TQ-S triple quadrupole mass spectrometer (MS/MS) running in positive ionisation mode with electrospray ionisation. A Waters BEH 50 × 2.1 mm, 1.7 µm C18 column, with a Waters BEH C18 5 × 2.1 mm, 1.7 µm guard cartridge maintained at 30 °C throughout the run, was used to chromatographically separate analytes injected at a volume of 2 µL over the six-minute run. Mobile phase A and B were delivered across the column at 0.4 mL/min over six minutes at varying gradient conditions (Table 7) to cause analyte separation.

Mobile phase A was MilliQ water and 1 mM ammonium fluoride in 95% water 5% methanol, whilst mobile phase B was 95% methanol 5% MilliQ water and 1 mM ammonium fluoride. Toxins were quantified via comparison with a reference certificate solution from sources detailed in Section 4.1. Working standard solutions were prepared by the appropriate dilutions of toxin stock in methanol and examples of the total ion chromatogram for each toxin in the level one standard are in Appendix A.

Table 8 lists the multiple reaction monitoring transitions, time and cone and collision voltage for each toxin. The limits of detection (LOD) and limits of quantitation (LOQ) for the three toxins present in this study are as follows: PnTx G had a LOD of 0.01 µg/kg and a LOQ 0.02 µg/kg, SPX1 had a LOD of 0.02 µg/kg and a LOQ of 0.06 µg/kg, and 20-Me-SPX-G had a LOD of 0.01 µg/kg and a LOQ of 0.03 µg/kg. The full list of LODs and LOQs for each compound and all other instrument and chromatographic criteria are as described in [82]. Any concentration above these amounts is considered as a detection of that toxin.

For the analysis of the hydrolysed and unhydrolysed samples, extracts were analysed using the same method and conditions used for regulatory lipophilic toxin monitoring, as stated in [97]. Sample analysis was performed using a Waters (Manchester, GB) Acquity I-Class UHPLC system coupled to a Xevo TQ-Abs triple quadrupole mass spectrometer (MS/MS) running in positive ionisation mode with electrospray ionisation. A Waters BEH 50 × 2.1 mm, 1.7 µm C18 column, with a Waters BEH C18 5 × 2.1 mm, 1.7 µm guard cartridge maintained at 30 °C throughout the run, was used to chromatographically separate analytes injected at a volume of 1 µL over the five-minute run. Mobile phase A and B were delivered across the column at 0.6 mL/min over five minutes at varying gradient conditions (Table 9) to cause analyte separation. Mobile phase A was MilliQ water and 0.1% ammonium hydroxide whilst mobile phase B was 90% acetonitrile, 10% MilliQ water and 0.1% ammonium hydroxide. The working standard solutions of PnTx G were prepared the same way as previously stated in Section 4.4. Multiple reaction monitoring transitions, time and cone and collision voltage for each toxin are the same as stated previously in Section 4.4.

LC-high resolution mass spectrometry (LC-HRMS) was used for the analysis of potential fatty acid esters. LC-HRMS was conducted using an Orbitrap Exploris 120 mass spectrometer coupled to a Vanquish UPLC system (Thermo Fisher, Hemel Hempstead, UK). Fatty acid esters were separated on an Acquity UPLC BEH C8 1.7 µm 2.1 × 50 mm column (Waters, Manchester, UK) maintained at 40 °C. Mobile phase A was 100% H_2_O, and B was 100% acetonitrile, both containing 0.1% formic acid. LC flow was maintained for 1 min at 25% B, followed by an increase to 100% B at 10 min, held for 2 min before returning to starting conditions at 12.5 min, with an overall cycle time of 15 min. The flow rate was 400 µL/min, and the injection volume was 5 µL for unhydrolyzed samples, and 6.25 µL for hydrolysed samples. LC effluent from 1 to 13 min was directed into the H-ESI source of the mass spectrometer which was operating in positive ion polarity with a voltage of 4500 V, sheath gas 50, sweep gas 1, ion transfer tube temp of 325 °C and vaporiser temp of 350 °C. The orbitrap was set to perform a full scan at a resolution of 120,000 with a scan range of *m*/*z* 600–1200, using an internal lock mass calibrant of *m*/*z* 203.0855. Quantitation of PnTx G esters was performed from the full scan data using the extracted ion chromatograms of the exact calculated masses detailed in [72]. Data were acquired using Xcalibur v4.5 (Thermofisher, Hemel Hempstead, UK) and processed using FreeStyle 1.8 (Thermofisher, Hemel Hempstead, UK). All fatty acid esters of PnTx G in this study were obtained without the use of known certified calibrated standards and based on detection and not quantification. All calculated exact masses were obtained from [72], and the relative mass error was calculated using these against our accurate measured mass, obtained from potential fatty acid ester peaks. Only peaks with a mass error < ±4 ppm were classed as potential peaks, and the absence of the same peak in the hydrolysed samples was used as confirmation.

### 4.5. Data Analysis

Prior to the analysis, all samples < LOQ were removed. CST samples are commercial samples with unknown origins and so were disregarded during interpretation. Shellfish have been grouped together by region rather than by exact location to maintain anonymity for the shellfish harvesters. Shapiro–Wilks test was used to assess the normality of the dataset. The non-parametric Kruskal–Wallis test was used to assess the difference between groups within each variable (region, month, season and species) against the concentration for each toxin. A Dunn pairwise non-parametric test was then used, following the Kruskal–Wallis results, as a multiple pairwise comparison post hoc analyses. A partial epsilon squared value with a 95% confidence interval was then used to measure the effect size of each group against each toxin. All analyses were performed using Excel [98] and R Statistical Software (v4.1.2; [99]) alongside the package’s car [100], dplyr [101], effect size [102], ggplot2 [103], ggstatsplot [104], janitor [105], paletteer [106] and readr [107]. ArcGIS Pro (v3.2.2; [108]) was used to create Appendix A.

## 5. Conclusions

Overall, this study has, for the first time, demonstrated the widespread presence of the CIs PnTx G, SPX1 and 20-Me-SPX-G in GB shellfish. Whilst there are currently no regulatory limits set for these toxins, some samples exceeded concentrations above the suggested human health guidance limits set by [23]. All samples contained concentrations below the suggested human health guidance limits set by [24], highlighting the importance of both the continued monitoring of concentrations and the requirement for toxicological and risk assessment studies.

## Figures and Tables

**Figure 1 marinedrugs-22-00556-f001:**
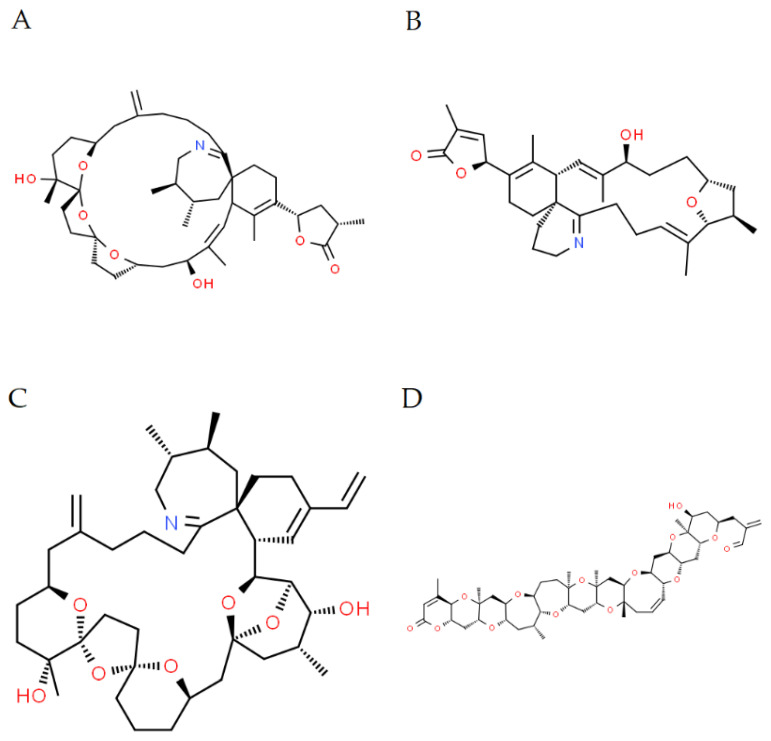
Example structure from each cyclic imine and brevetoxin group. (**A**) SPX1 [78]; (**B**) Gymnodimine A [79]; (**C**) PnTx G [80]; (**D**) Brevetoxin-2 [81].

**Figure 2 marinedrugs-22-00556-f002:**
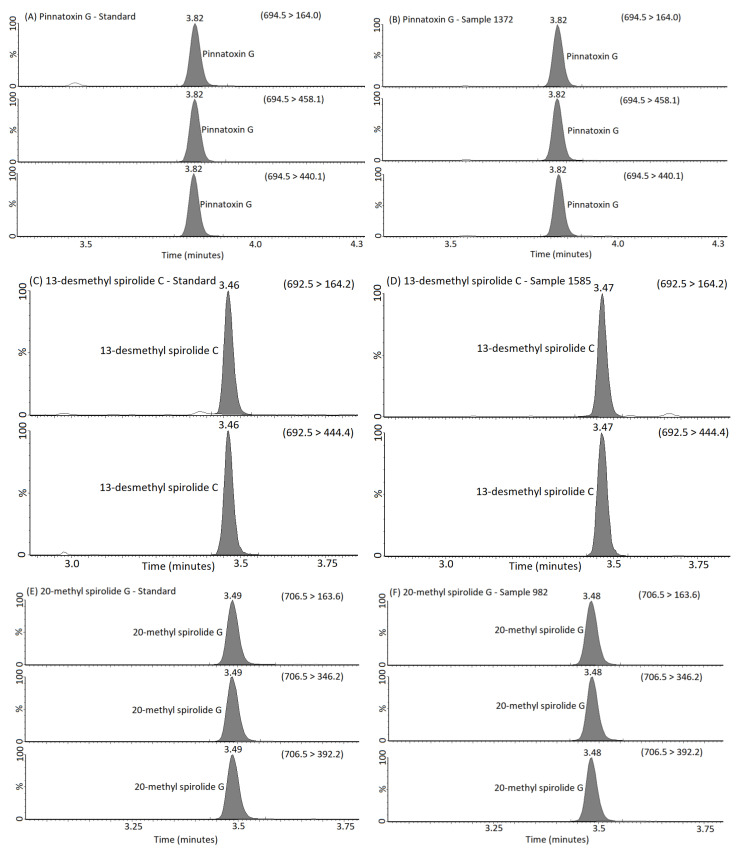
MRM chromatograms obtained following the analysis of (**A**) PnTx G standard; (**B**) sample 1372 containing the highest PnTx G concentration; (**C**) SPX1 standard; (**D**) sample 1585 containing the highest SPX1 concentration; (**E**) 20-Me-SPX-G standard; (**F**) sample 982 containing the highest concentration of 20-Me-SPX-G. Primary and secondary MRM transitions and retention times are labelled.

**Figure 3 marinedrugs-22-00556-f003:**
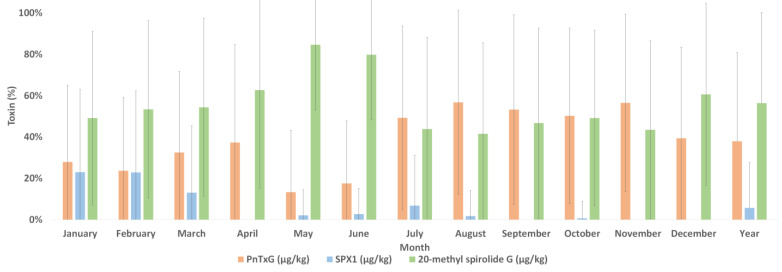
Mean toxin profile (concentration of each toxin relative to the total in each sample) ± SD for samples received each month from January–December 2018 and for the full year.

**Figure 4 marinedrugs-22-00556-f004:**
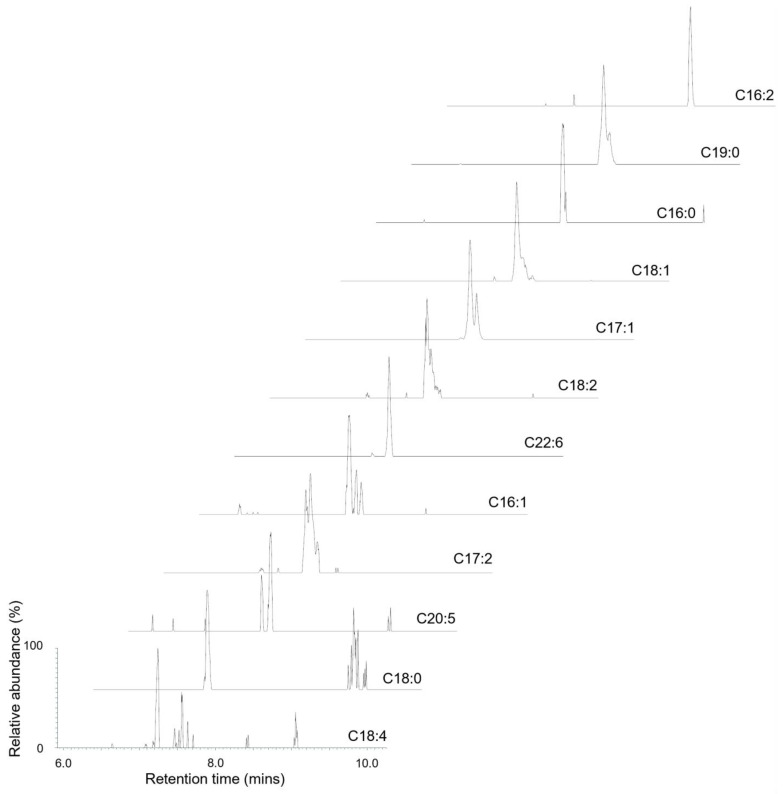
LC-HRMS analysis of fatty acid esters in two mussel samples containing the highest difference in concentration (Appendix A; samples three and eight).

**Table 1 marinedrugs-22-00556-t001:** Summary of cyclic imine and brevetoxin analogues monitored in this study.

Spirolides	Gymnodimines	Pinnatoxins	Brevetoxins
13-desmethyl Spirolide C (SPX1)	Gymnodimine (GYM)	Pinnatoxin A (PnTx A)	Brevetoxin-2 (PbTX2)
13, 19 didesmethyl spirolide C (13,19-didesMe-SPX-C)	12-methyl gymnodimine (12-Me-GYM)	Pinnatoxin D (PnTx D)	Brevetoxin-3 (PbTX3)
20-Methyl Spirolide G (20-Me-SPX-G)		Pinnatoxin E (PnTx E)	Brevetoxin-B2 (BTX B2)
27-oxo-13,19-didesmethyl Spirolide C (27-oxo-13,19-didesMe-SPX-C)		Pinnatoxin F (PnTx F)	S-desoxy brevetoxin-B2 (S-desoxy-BTX-B2)
27-hydroxy-13,19-didesmethyl Spirolide C (27-OH-13,19-didesMe-SPX-C)		Pinnatoxin G (PnTx G)	Brevetoxin-B4 (BTX-B4)
27-hydroxy-13-desmethyl Spirolide C (27-OH-13-desMe-SPX-C)		Pinnatoxin H (PnTx H)	Brevetoxin-B5 (BTX B5)
		Portimine	

**Table 2 marinedrugs-22-00556-t002:** Comparison of toxin concentrations within and between geographic regions with a positive detection in 2018. Rows within an area denote different locations within that geographic region. Geographic regions are visualised within Appendix A; nd denotes not detected.

Geographic Regions	Mean Toxin Concentration (µg/kg) (SD)			Total Sampled in Location
	PnTx G	SPX1	20-MSPX G	
Central–North Scotland	3.4 (2.4)	0.8 (1.1)	7.6 (8.2)	115
Northeast Scotland	9.6 (8.4)	0.7 (0.7)	5.8 (5.4)	581
Northwest Scotland	3.5 (3.2)	0.2 (0.4)	4.6 (4.8)	202
	4.0 (4.6)	0.6 (0.2)	3.6 (2.3)	37
	1.9 (3.0)	0.9 (0.7)	6.2 (6.0)	142
	4.5 (2.9)	0.1 (nd)	8.2 (8.0)	48
	2.7 (2.6)	0.7 (0.6)	6.3 (6.3)	81
Central–West Scotland	14.2 (14.9)	0.3 (0.4)	4.4 (4.7)	490
	nd	1.0 (1.1)	3.6 (3.0)	31
Central–East Scotland	nd	3.4 (3.3)	3.1 (2.4)	71
Southeast Scotland	nd	4.9 (nd)	1.0 (nd)	9
Southwest Scotland	nd	1.8 (nd)	5.0 (3.1)	22
	nd	1.1 (0.2)	nd	20
North Wales	nd	0.6 (0.6)	nd	10
	nd	0.9 (0.6)	nd	26
	nd	0.6 (0.2)	nd	11
South Wales	nd	0.5 (0.2)	0.4 (0.7)	16
	nd	1.3 (0.8)	nd	22
	7.6 (7.4)	nd	nd	9
Northeast England	nd	0.6 (0.7)	23.2 (14.4)	14
Northwest England	nd	0.9 (0.5)	nd	11
	nd	1.2 (1.3)	2.4 (3.3)	24
	nd	1.5 (nd)	nd	3
	nd	3.0 (nd)	nd	4
	nd	0.6 (0.6)	nd	12
	nd	nd	4.0 (nd)	9
Central–East England	nd	1.0 (1.2)	nd	11
	nd	0.1 (0.0)	nd	26
Central–South England	nd	0.8 (nd)	nd	12
Southeast England	nd	1.0 (0.0)	nd	10
	nd	0.5 (0.4)	nd	36
	nd	1.7 (1.8)	nd	31
	nd	0.7 (0.8)	nd	36
	nd	1.1 (0.7)	nd	24
	nd	0.4 (0.2)	nd	10
Southwest England	nd	13.4 (nd)	1.4 (1.0)	15
	8.5 (11.2)	2.7 (3.5)	10.3 (10.3)	163
	nd	0.4 (0.3)	nd	12
	nd	nd	0.4 (nd)	1
	nd	2.4 (3.7)	3.0 (3.2)	15
	nd	1.3 (nd)	nd	12
	nd	1.2 (1.3)	3.0 (2.3)	23
	nd	1.2 (1.7)	nd	24
	nd	0.5 (0.6)	nd	24
	nd	0.7 (nd)	1.2 (1.0)	36
	nd	0.1 (nd)	0.3 (0.2)	20
	nd	0.3 (0.4)	3.5 (3.0)	13
	nd	0.2 (0.1)	0.5 (0.6)	12
Unknown	1.4 (1.3)	0.6 (0.5)	3.9 (1.8)	62

**Table 3 marinedrugs-22-00556-t003:** Comparison of temporal (monthly) toxin concentrations in 2018; nd denotes not detected.

		Months											
		January	February	March	April	May	June	July	August	September	October	November	December
Total sampled each month		136	141	136	203	235	250	338	298	293	296	211	134
PnTxG (µg/kg)	Mean	6.18	5.50	6.24	3.96	4.73	10.61	11.52	11.65	9.98	10.53	9.19	7.27
	SD	6.15	6.58	5.43	3.51	4.96	13.72	14.47	11.81	8.66	9.43	8.48	5.90
	Count	47	42	38	26	32	64	107	91	93	96	65	28
	% pos (month)	34.56	29.79	27.94	12.81	13.62	25.60	31.66	30.54	31.74	32.43	30.81	20.90
SPX1 (µg/kg)	Mean	0.67	0.66	0.49	nd	1.09	1.79	3.31	3.77	nd	0.20	nd	nd
	SD	0.68	0.66	0.50	nd	1.03	2.57	3.93	3.78	nd	0.00	nd	nd
	Count	50	49	25	nd	11	22	17	3	nd	1	nd	nd
	% pos (month)	36.76	34.75	18.38	nd	4.68	8.80	5.03	1.01	nd	0.34	nd	nd
20-Me-SPX-G (µg/kg)	Mean	4.82	4.55	4.52	4.31	8.54	7.26	6.34	4.84	4.83	4.90	5.73	5.92
	SD	4.53	4.60	5.10	6.22	8.34	7.95	6.35	3.58	5.53	5.07	7.13	8.81
	Count	79	87	67	44	128	197	107	84	101	104	60	43
	% pos (month)	58.09	61.70	49.26	21.67	54.47	78.80	31.66	28.19	34.47	35.14	28.44	32.09

**Table 4 marinedrugs-22-00556-t004:** Comparison of toxin concentrations within and between species in 2018; nd denotes not detected.

Species	Toxin	Mean Concentration (µg/kg) (SD)	% of Species Positive to Total Sampled	Maximum Concentration (µg/kg)
Cockles	PnTx G	4.0 (4.6)	9.8	14
	SPX1	0.9 (0.8)	12.3	3
	20-Me-SPX-G	2.8 (2.4)	15.6	7.7
Hard clams	PnTx G	nd	0.0	nd
	SPX1	0.6 (0.3)	8.3	0.8
	20-Me-SPX-G	nd	0.0	nd
King scallop	PnTx G	nd	0.0	nd
	SPX1	0.6 (0.3)	5.0	0.8
	20-Me-SPX-G	nd	0.0	nd
Manila clams	PnTx G	nd	0.0	nd
	SPX1	0.4 (0.3)	25.0	0.6
	20-Me-SPX-G	nd	0.0	nd
Mussels	PnTx G	9.4 (10.4)	41.0	85.4
	SPX1	1.1 (2.1)	5.5	13.4
	20-Me-SPX-G	5.1 (5.4)	40.0	43.9
Native oysters (*Ostrea edulis*)	PnTx G	nd	0.0	nd
	SPX1	1.3 (1.3)	11.1	3.2
	20-Me-SPX-G	22.5 (11.6)	38.9	41.5
Pacific oysters	PnTx G	4.6 (5.7)	1.9	17.4
	SPX1	0.8 (1.2)	7.9	7.8
	20-Me-SPX-G	6.6 (7.1)	63.2	51.4
Razor clams (*Ensis* spp. and *Solen* spp.)	PnTx G	nd	0.0	nd
	SPX1	3.6 (3.0)	9.1	8.4
	20-Me-SPX-G	1.6 (1.4)	5.2	3.5
Surf clams (*Spisula* spp.)	PnTx G	nd	0.0	nd
	SPX1	1.9 (1.7)	7.1	3.4
	20-Me-SPX-G	3.9 (2.8)	11.9	7.1

**Table 5 marinedrugs-22-00556-t005:** Comparison of effect sizes on toxins obtained by Kruskal–Wallis with Dunn pairwise output. * Indicates a statistically significant relationship between variables and the toxin concentrations detected.

	PnTx G	SPX1	20-Me-SPX-G
Location	Large *	Large	Medium *
Month	Medium *	Medium *	Small *
Season	Small *	Medium *	Small *
Species	Very Small *	Medium	Small *

**Table 6 marinedrugs-22-00556-t006:** Summary of PnTx G fatty acid esters detected in two mussel samples containing the highest difference in concentration (Appendix A; samples three and eight), sampled seven days apart from the same site in July 2023. ^x^ Represents fatty acid esters only found in the mussel sample collected in the first week. ^s^ Represents fatty acid esters only found in the mussel sample collected on the second week. N/A denotes not applicable.

Fatty Acid	Calculated Exact Mass	Measured Accurate Mass	Mass Error (Δ ppm)	RT (min)	Peak Area	% of Total Peak Area
14:0	904.6661	N/A	N/A	N/A	N/A	N/A
15:0	918.6817	N/A	N/A	N/A	N/A	N/A
16:2	928.6661	928.6653	−0.86	9.13	3.02 × 10^6^	1.34%
16:1	930.6817	930.6807	−1.07	7.91	4.02 × 10^6^	1.78%
16:0	932.6974	932.6987	1.39	8.38	1.74 × 10^6^	0.77%
17:2	942.6817	942.6816	−0.11	7.86	1.42 × 10^7^	6.29%
17:1	944.6974	944.6968	−0.64	8.09	1.22 × 10^8^	54.25%
17:0	946.7130	N/A	N/A	N/A	N/A	N/A
18:4	952.6661	952.6642	−1.99	7.24	1.47 × 10^6^	0.65%
18:3	954.6817	N/A	N/A	N/A	N/A	N/A
18:2	956.6974	956.6971	−0.31	7.99	9.55 × 10^6^	4.23%
18:1	958.7130	958.7132	0.21	8.24	2.54 × 10^7^	11.26%
18:0 ^x^	960.7287	960.7288	0.10	7.43	1.51 × 10^6^	0.67%
19:0	972.7287	972.7288	0.10	8.46	2.42 × 10^7^	10.73%
20:5 ^x^	978.6817	978.6821	0.41	7.8	1.29 × 10^6^	0.57%
20:4	980.6974	N/A	N/A	N/A	N/A	N/A
20:3	982.7130	N/A	N/A	N/A	N/A	N/A
20:2	984.7287	N/A	N/A	N/A	N/A	N/A
20:1	986.7443	N/A	N/A	N/A	N/A	N/A
22:6	1004.6974	1004.6967	−0.70	7.96	7.85 × 10^6^	3.48%
22:5	1006.7130	1006.7097	−3.28	7.98	8.29 × 10^6^	3.67%
22:4	1008.7287	N/A	N/A	N/A	N/A	N/A
22:3	1010.7443	N/A	N/A	N/A	N/A	N/A
22:2 ^s^	1012.7510	1012.7547	3.65	6.95	6.88 × 10^5^	0.79%
24:6	1032.7287	N/A	N/A	N/A	N/A	N/A
24:5	1034.7443	N/A	N/A	N/A	N/A	N/A

**Table 7 marinedrugs-22-00556-t007:** Mobile phase A and B gradient conditions on LC.

Time (min)	Mobile Phase A (%)	Mobile Phase B (%)
0	95	5
0.3	95	5
1	80	20
1.5	65	35
2	50	50
2.5	35	65
3	20	80
3.5	10	90
4	5	95
5.5	0	100
6	95	5

**Table 8 marinedrugs-22-00556-t008:** Multiple reaction monitoring transitions, time and cone and collision voltage used in positive ionisation mode during LC-MS/MS analyses. Each consecutive cone and collision voltage number correlates to each consecutive primary quantitative product ion.

Toxin Group	Compound	MRM Transition (*m/z*)	Time (min)	Cone (V)	Collision (eV)
Spirolides	SPX1	692.5 > 164.2, 444.4	2.4–4.4	40, 40	45, 40
	13,19-didesMe-SPX-C	678.5 > 164.1, 430.1	2.8–4.4	40, 40	50, 35
	20-Me-SPX-G	706.5 > 163.6, 346.2, 392.2	3.0–4.4	40, 40, 40	50, 35, 30
	27-oxo-13,19-didesMe-SPX-C	692.42 > 178.12, 444.27	0.8–6	40, 40	40, 40
	27-OH-13,19-didesMe-SPX-C	694.5 > 180.14, 464.3	2.7–3.95	40, 40	40, 40
	27-OH-13-desMe-SPX-C	708.45 > 180.14, 478.32	0.8–6.0	40, 40	40, 40
Gymnodimines	GYM	508.4 > 136.1, 162.2	2.5–3.6	35, 35	38,38
	12-Me-GYM	522.7 > 120.5, 135.0, 246.0, 300.0	2.7–4.4	40, 40, 40, 40	40, 40, 35, 35
PnTxs	PnTx A	712.5 > 164.1, 458.3	0.8–6.0	40, 40	40, 40
	PnTx D	782.0 > 164.0	1.0–6.0	40	55
	PnTx E	784.2 > 164.0, 446.2, 488.17	2.5–3.8	40, 40, 40	55, 45, 45
	PnTx F	766.4 > 164, 446.2, 488.17	3.0–4.4	40, 40, 40	55, 45, 45
	PnTx G	694.5 > 164.0, 440.1, 458.1	3.3–5.0	40, 40, 40	55, 45, 50
	PnTx H	708.0 > 164.0	1.0–6.0	40	55
	Portimine	402.2 > 164	1.0–6.0	40	55
Brevetoxins	PbTx-1	867.2 > 221.0, 385.0, 611.0	4.0–5.0	60, 60, 60	30, 30, 30
	PbTX2	895.5 > 319.2, 877.5 and 912.5 > 319.2	4.0–5.1	60, 60, 60	30, 30, 30
	PbTX3	897.5 > 129.0, 725.5	4.0–5.0	60, 60	30, 30
	BTX B2	1034.5 > 220.0, 911.0, 929.0, 947.0	3.0–5.0	40, 40, 40, 40	40, 40, 40, 40
	S-desoxy-BTX-B2	1018.6 > 204.1, 248.2	3.6–5.0	40, 40	40, 40
	BTX-B4	1272.7 > 326.2, 911.3, 929.4	0.8–6.0	80, 80, 80	35, 35, 35
	BTX B5	911.5 > 753.0, 839.0, 873.5 and 928.5 > 875.5 and 933.5 > 875.5	3.0–5.0	60, 60, 60, 60, 60, 60	20, 20, 20, 20, 20, 20

**Table 9 marinedrugs-22-00556-t009:** Mobile phase A and B conditions on LC-MS/MS for PnTx G unhydrolysed vs. hydrolysed samples.

Time (min)	Mobile Phase A (%)	Mobile Phase B (%)
0	85	15
2	60	40
3	0	100
3.5	0	100
3.6	85	15
5	85	15

## Data Availability

Data are contained within the article and the Appendix A.

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
