# Peer review of "Detection of the Cyclic Imines Pinnatoxin G, 13-Desmethyl Spirolide C and 20-Methyl Spirolide G in Bivalve Molluscs from Great Britain"

_marinedrugs, 2024, doi:10.3390/md22120556_

Round 1
Reviewer 1 Report
Comments and Suggestions for Authors
The research presents a comprehensive overview of cyclic imines along the coasts of Great Britain during 2018.
I have a few remarks for improvement:
Fig. 2 – is not readable, the file was too much compressed and toxin names and toxin masses are not clear to the reader.
Line 152- the « average toxin profile across all samples», is this the yearly average? I suggest putting the second and third sentences (line 153-154), before this one, on line 152. Also, this section is too short; it can be further elaborated. When was the lowest percentage of PnTxG detected in the profile?
Line 181- are the «locations», the classified production areas in the UK? I suggest that table 2 requires a geographic map, maybe as a the supplementary file, for better comprehension of these ‘regions’ and ‘locations’.
Line 427-when discussing results from Ingril lagoon, and similar areas, please put the remark that this is located in the Mediterranean, not the Atlantic, that presents much warmer water temperatures, etc.
Line 537-same comment as above. Were BTX toxins only detected in the French Mediterranean coast? Were BTX ever been detected in any Atlantic coasts of Europe?
Author Response
Comment 1: Fig. 2 – is not readable, the file was too much compressed and toxin names and toxin masses are not clear to the reader.
Response 1: Thank you for the comment, I have now separated the chromatograms into their original files so that they aren't compressed into the one file and made the text bigger and clearer. They now take up a larger space on the manuscript but this should hopefully stop the compression. I have updated the figure and also increased the size of the text.
Comment 2: Line 152- the « average toxin profile across all samples», is this the yearly average? I suggest putting the second and third sentences (line 153-154), before this one, on line 152. Also, this section is too short; it can be further elaborated. When was the lowest percentage of PnTxG detected in the profile?
Response 2: Thank you for the comment, yes this is the yearly average, have added in 'annual' to indicate this alongside putting lines 153-154 before 152. I have also elaborated further by including when the highest and lowest percentage of each toxin was detected in the profile.
Comment 3: Line 181- are the «locations», the classified production areas in the UK? I suggest that table 2 requires a geographic map, maybe as a the supplementary file, for better comprehension of these ‘regions’ and ‘locations’.
Response 3: Yes the locations are classified production areas, but unfortunately the permission to use the samples was on the premise that we gave anonymity to the harvesters upon publication. Hence why the production areas were not included and thus why we viewed the data in a much broader context (regions). I have included a map of the UK in the supplementary materials (Figure 11) that includes labels for the geographic regions, however I will unfortunately not be able to include the details on locations.
Comment 4: Line 427-when discussing results from Ingril lagoon, and similar areas, please put the remark that this is located in the Mediterranean, not the Atlantic, that presents much warmer water temperatures, etc.
Response 4: Thank you for this comment, I have added in this remark to Ingril lagoon on line 427 and have also elaborated further on water bodies when discussing other areas within the discussion.
Comment 5: Line 537-same comment as above. Were BTX toxins only detected in the French Mediterranean coast? Were BTX ever been detected in any Atlantic coasts of Europe?
Response 5: Thank you again for recognising this comment, BTXs (BTX-2, BTX-3) were only found on the Mediterranean coast and not the Atlantic coast whilst GYM-A was present at both the Atlantic and Mediterranean coast. We have now addressed this in the text. And as above, have elaborated further on water bodies when discussing other areas within the discussion.
Reviewer 2 Report
Comments and Suggestions for Authors
The paper is a good study which would add valuable knowledge into the literature. However, at present the quality of the writing is poor. Spelling mistakes, abbreviation errors, etc are numerous but could be easily fixed. Of more concern is the repetitive nature of the paper which makes it difficult to read. The results of the study are all in tables but a lot are also listed in the text. I found the discussion to be the major problem as it was very long containing a lot of data out of the results section. Perhaps separate sections for temporal trends, species etc are not needed as the story needs to be pulled together in a much more concise manner.
I have attached a file with my more specific comments

Comments on the Quality of English Language
I think that I have addressed this in my comments above
Reviewer 3 Report
Comments and Suggestions for Authors
Nice, very comprehensive manuscript. Only suggestion would be to include in Discussion some comparison of the results with those of other studies done in the north Atlantic waters.
Author Response
Comment 1: Only suggestion would be to include in Discussion some comparison of the results with those of other studies done in the north Atlantic waters.
Response 1: Thank you for your comments, we have gone through the discussion and defined geographic locations of studies that have been used already in creating comparisons with the current study. In the revised manuscript this corresponds to lines:
- 338 - 340 + 345
- 347 - 349 + 355 - 357
- 362 - 363
- 374 - 378
- 378 - 380
- 382 - 383
- 389 + 390
- 422 - 425
- 440 - 441
- 446 - 450
- 454 - 460
- 470
- 478
- 480
- 484
- 491 - 492
- 508 - 509
- 526
- 530
- 549 - 553
- 569 - 571
- 584 - 588
- 620
Round 2
Reviewer 2 Report
Comments and Suggestions for Authors
Please see attached for my comments

Comments on the Quality of English Language
This is better than the first version but I still think it could be significantly improved. I have addressed this in my comments.
Author Response
Thank you for your review and comments, these have now been addressed.
Please see the attachment.
